



**Observation of atmospheric peroxides during Wangdu Campaign**
**2014 at a rural site in the North China Plain**
**Yin Wang, Zhongming Chen, Qinqin Wu, Hao Liang, Liubin Huang, Huan Li,**
**Keding Lu, Yusheng Wu, Huabin Dong, Limin Zeng, and Yuanhang Zhang**
State Key Laboratory of Environmental Simulation and Pollution Control, College of
Environmental Sciences and Engineering, Peking University, Beijing 100871, China
*Correspondence to:* Zhongming Chen (zmchen@pku.edu.cn)
**Abstract**
Measurements of atmospheric peroxides were made during Wangdu Campaign 2014 at
Wangdu, a rural site in the North China Plain (NCP) in summer 2014. The predominant
peroxides were detected to be hydrogen peroxide ($H_2O_2$), methyl hydroperoxide (MHP)
and peroxyacetic acid (PAA). The observed $H_2O_2$ reached up to 11.3 ppbv, which was
the highest value compared with previous observations in China at summer time. A box
model simulation based on the Master Chemical Mechanism and constrained by the
simultaneous observations of physical parameters and chemical species was performed
to explore the chemical budget of atmospheric peroxides. Photochemical oxidation of
alkenes was found to be the major secondary formation pathway of atmospheric
peroxides, while contributions from alkanes and aromatics were of minor importance.
The comparison of modelled and measured peroxide concentrations revealed an
underestimation during biomass burning events and an overestimation on haze days,
which were ascribed to the direct production of peroxides from biomass burning and
the heterogeneous uptake of peroxides by aerosols, respectively. The strengths of the
primary emissions from biomass burning were on the same order of the known
secondary production rates of atmospheric peroxides during the biomass burning events.
The heterogeneous process on aerosol particles was suggested to be the predominant
sink for atmospheric peroxides. The atmospheric lifetime of peroxides on haze days in
summer in the NCP was about 2–3 hours, which is in good agreement with the



laboratory studies. Further comprehensive investigations are necessary to better
understand the impact of biomass burning and heterogeneous uptake on the
concentration of peroxides in the atmosphere.
**1 Introduction**
Atmospheric peroxides, including hydrogen peroxide ($H_2O_2$) and organic peroxides
(ROOH), are vital oxidants present in the gaseous, aqueous and particulate phase in the
atmospheric chemical processes. They serve as temporary reservoirs for $HO_x$ radicals,
contributing to the atmospheric oxidation capacity (Reeves and Penkett, 2003).
Peroxides also participate in the conversion of S(IV) to S(VI) in the aqueous phase,
leading to the acid precipitation and the formation of secondary sulfate ($SO_4^{2-}$) aerosols
in the troposphere (Calvert et al., 1985; Stein and Saylor, 2012). Furthermore,
atmospheric peroxides are considered as the key components of secondary organic
aerosol (SOA), which play a significant role in the formation and duration of haze
pollution (Kroll and Seinfeld, 2008; Ziemann and Atkinson, 2012; Li et al., 2016). In
addition, it has been suggested that atmospheric peroxides are toxic to ecosystem and
may be the critical pollutants of forest decline (Hellpointner and Gäb, 1989; Chen et al.,
2010). More importantly, peroxides in the particle phase have been found to act as
reactive oxygen species (ROS) and result in adverse influence on human health (Ayres
et al., 2008).
The concentrations of atmospheric peroxides are determined by their production and
destruction. The known formation pathways of peroxides in the atmosphere are primary
emissions, for instance, biomass burning (Lee et al., 1997, 1998; Yokelson et al., 2009),
and secondary sources such as peroxy radical self/cross reactions and the ozonolysis of
unsaturated volatile organic compounds (VOCs), as shown in Reaction (R1, R2) and
(R3, R4), respectively (Hewitt and Kok, 1991; Neeb et al., 1997; Sauer et al., 2001;
Chao et al., 2015; Winiberg et al., 2016). Additionally, atmospheric aqueous reactions
in the bulk solution or on the surface of wet particles coupled with subsequent release
to the gas phase could also generate peroxides in the troposphere (Wang et al., 2012;
Liang et al., 2013a; Zhao et al., 2013a). The typical removal pathways of peroxides in



the atmosphere are photolysis (R5, R6), reaction with OH radicals (R7, R8) and
physical deposition (Atkinson et al., 2006; Sander et al., 2011; Nguyen et al., 2015).
Heterogeneous uptake by atmospheric aerosols is recognized as another significant sink
for peroxides in the troposphere, especially in dusty and polluted urban areas (Zhao et
al., 2013b; Wu et al., 2015).

$$HO_2 + HO_2 \rightarrow H_2O_2 + O_2 \tag{R1}$$

$$RO_2 + HO_2 \rightarrow ROOH + O_2 \tag{R2}$$

$$RCH=CH_2 + O_3 \rightarrow RCHOO + HCHO \tag{R3}$$

$$RCHOO + H_2O \rightarrow RCH(OH)OOH \tag{R4}$$

$$H_2O_2 + h\nu \rightarrow OH + OH \tag{R5}$$

$$ROOH + h\nu \rightarrow RO + OH \tag{R6}$$

$$H_2O_2 + OH \rightarrow HO_2 + H_2O \tag{R7}$$

$$ROOH + OH \rightarrow RO_2 + H_2O \tag{R8}$$

In the past years, a number of field observations, laboratory studies and modelling
research have been carried out to investigate the abundance and behavior of peroxides
in the atmosphere (Chen et al., 2008; Mao et al., 2010; Huang et al., 2013; Liang et al.,
2013a; Sarwar et al., 2013; Epstein et al., 2014; Fischer et al., 2015; Khan et al., 2015).
Hydrogen peroxide ($H_2O_2$), hydroxymethyl hydroperoxide (HMHP, $HOCH_2OOH$),
methyl hydroperoxide (MHP, $CH_3OOH$) and peroxyacetic acid (PAA, $CH_3C(O)OOH$)
are generally determined to be the principal peroxide compounds in the troposphere
with their concentrations ranging from pptv (parts per trillion by volume) to ppbv (parts
per billion by volume) (Lee et al., 2000; He et al., 2010; Zhang et al., 2010, 2012).
However, to date, there have been limited studies concerned with atmospheric
peroxides in the regions primarily affected by anthropogenic sources such as the North
China Plain (NCP), which is a typical region with frequent biomass burning and
suffering from serious haze pollution in China (Tao et al., 2012; Huang et al., 2014).
Few numerical simulations focused on atmospheric peroxides in the NCP are conducted
to examine whether the models can reproduce the observations of peroxides (Liang et
al., 2013a). The impact of biomass burning and high aerosol loading on the atmospheric
chemistry of peroxides over such a polluted region is poorly understood. Therefore, this



work was carried out in order to make an endeavor to fill in these research gaps.
In this study, we present a novel dataset of atmospheric speciated peroxides and
explore their atmospheric chemistry at a rural site, Wangdu, which represents regional
air pollution conditions of the NCP during Wangdu Campaign 2014. Given the diversity
of emission sources and chemical transformation of atmospheric peroxides over this
region, it is challenging to analyze the phenomena and understand the primary emission
and secondary formation of peroxides in the atmosphere during this field observation.
However, with the continuous measurements of atmospheric peroxides, physical
parameters and other chemical species performed simultaneously, a quantitative
assessment of the budget of atmospheric peroxides can be carried out employing the
zero-dimensional model based on Master Chemical Mechanism (MCM) and
constrained by observed meteorological parameters and trace gases, which provides a
good opportunity to comprehensively facilitate our knowledge of the chemistry of
atmospheric peroxides in the NCP. As far as we know, this is the first study to test
whether current atmospheric peroxides related chemistry could explain the field
observation in the rural area of the NCP. Through the comparison between measurement
and simulation, our aim is to investigate the role of biomass burning and heterogeneous
uptake on aerosols in the concentration of atmospheric peroxides, which helps to
develop more robust mechanism in the model.
**2 Experiments**
**2.1 Measurement site**
Measurements of atmospheric peroxides were performed at Wangdu site (38.66 °N,
115.20 °E) in Baoding city, Hebei Province, a rural supersite for the Wangdu Campaign
2014 situated in the northwest of the NCP, about 200 km southwest of the mega-city
Beijing. The surrounding regions of Wangdu site are mainly agricultural fields. There
are almost no industries near this site. During the summer season, the air pollution is
caused by the primary emission from biomass burning and secondary formation
including photochemical and heterogeneous processes. The instruments were placed in
a container with the sampling inlet approximately 5 m above the ground. The





continuous observation of atmospheric peroxides was conducted from 4 June to 7 July

109 2014.

## 2.2 Measurement methods

### 2.2.1 Measurement method for atmospheric peroxides

Atmospheric peroxide concentrations were investigated by an automated on-site high
performance liquid chromatography (HPLC) with post-column enzyme derivatization
and detected by fluorescence spectroscopy. Air samples were pumped through a glass
scrubbing coil maintained at a controlled temperature of about $4\,℃$ to collect the
peroxides in the atmosphere. The flow rate of air samples was set to be 2.7 standard L
$min^{-1}$. The stripping solution, $5\times10^{-3}$ M $H_3PO_4$ in water was delivered into the
scrubbing coil collector. The flow rate of stripping solution was set to be 0.2 mL $min^{-1}$.
Once the air samples mixed with the stripping solution in the collector, the mixture was
carried by the mobile phase containing $5\times10^{-3}$ M $H_3PO_4$ at 0.5 mL $min^{-1}$ and injected
into HPLC. The peroxide components were separated after the mixture passed through
HPLC column. With the catalysis of Hemin at $\sim40\,℃$, the derivatization reaction
between peroxide components and para-hydroxyphenylacetic acid (PHPAA) produced
the fluorescent matter that can be quantified by fluorescence detector. In this work,
atmospheric peroxides were measured every 20 min. The collection efficiencies for
hydrogen peroxide and organic peroxides were determined to be 100% and 85%,
respectively. The detection limit of peroxides in the gas phase was about 10 pptv.
The interference of $SO_2$ on the sampling was estimated using the theoretical
thermodynamic and kinetic analysis presented in Hua et al. (2008). Considering the rate
constant for reaction between peroxides and S(IV) reported by Ervens et al. (2003) and
the mean level of $SO_2$ was $7.0\pm7.0$ ppbv during the campaign, the negative artifact
caused by $SO_2$ interference for peroxides was calculated to be less than 15%. The
influence of ambient relative humidity (RH) on the measurement of atmospheric
peroxides was calculated following the method introduced by Liang et al. (2013b). The
change of the concentration of atmospheric peroxides after this calibration is less than
10%. Here, we did not correct the observational data for any artifacts due to the





uncertainties from the theoretical estimation of peroxides loss that possibly result in
new errors. The uncertainty of our observational data is estimated to be ~15%. Further
details about our measurement method for atmospheric peroxides can be obtained from
Hua et al. (2008).

**2.2.2 Measurement methods for other pollutants and parameters**

During Wangdu Campaign 2014, $SO_2$, CO, $NO/NO_2$ and $O_3$ were measured
concurrently at this supersite using a suite of commercial instruments (Thermo 43i, 42i,
48i and 49i). HONO was measured every 2 min with a LOng Path Absorption
Photometer (LOPAP) (Liu et al., 2016). $C_2$–$C_{10}$ non-methane hydrocarbons (NMHCs)
were analyzed with a time resolution of 60 min by a custom-built online VOC analyzer
using automated gas chromatography (GC) coupled with flame ionization detector (FID)
or mass spectrometry (MS) technique (Wang et al., 2014). OH and $HO_2$ radicals were
measured by laser-induced fluorescence (LIF) spectroscopy. Size distributions of
aerosols ($PM_{10}$) were determined every 10 min with a Twin Differential Mobility
Particle Sizer (TDMPS) and an Aerodynamic Particle Sizer (APS) to calculate dry
particle surface area concentrations ($S_a$). Hygroscopic growth factor, $f$ (RH), which is
defined as the ratio of scattering coefficient for ambient aerosol to scattering coefficient
for dry aerosol, was derived from the integrating nephelometer (Liu, 2015).
Measurements of the mass concentration of $PM_{2.5}$ were obtained by TEOM 1400A
analyzer. Photolysis frequencies were derived from a spectro-radiometer (Bohn et al.,
2008). Meteorological parameters including ambient temperature, relative humidity
(RH), pressure, wind speed, wind direction and rainfall were monitored continuously
by a weather station. The uncertainties (1σ) in these measurements are estimated as 5%
for NO, $O_3$, and CO, 10% for $H_2O$, $NO_2$, HONO, NMHCs, and solar radiation, and 20%
for $S_a$.

**2.3 Model description**

A zero-dimensional box model using a near-explicit mechanism, MCM Version 3.3.1
(http://mcm.leeds.ac.uk/MCM/) (Jenkin et al., 1997, 2003; Saunders et al., 2003; Jenkin
et al., 2015) was employed to examine the influence of biomass burning and



heterogeneous uptake on the budget of atmospheric peroxides. MCMv3.3.1 describes
the degradation of 143 VOCs, leading to about 5800 species and 17000 reactions. In
the current study, we extracted a subset of MCMv3.3.1 containing the reactions of
atmospheric oxidants with measured VOCs and subsequent chemical products.
Measurements of $NO/NO_2$, CO, $O_3$, HONO, NMHCs, temperature, pressure and $H_2O$
were used as inputs to constrain the model calculations. The model ran with a 5-min
time step and a spin-up time of 2 days to reach a steady state. Photolysis frequencies
were calculated by the Tropospheric Ultraviolet and Visible (TUV, version 5.2) model
(Madronich, 2002), and further rescaled with the measured $j(NO_2)$. Dry deposition
velocities of trace gases in our box model were parameterized as $V_d/h$ (Seinfeld and
Pandis, 2006), where $V_d$ is the dry deposition rate of species and $h$ is the height of
planetary boundary layer (PBL). Dry deposition rates of $HNO_3$, PANs, organic nitrates,
$H_2O_2$, organic peroxides and aldehydes incorporated in the model were set as $2.0 \times 10^{-5}$
$s^{-1}$, $5.0 \times 10^{-6}$ $s^{-1}$, $1.0 \times 10^{-5}$ $s^{-1}$, $1.0 \times 10^{-5}$ $s^{-1}$, $5.0 \times 10^{-6}$ $s^{-1}$ and $1.0 \times 10^{-5}$ $s^{-1}$, respectively
at the PBL height of 1 km (Zhang et al., 2003; Emmerson et al., 2007; Lu et al., 2012;
Guo et al., 2014; Li et al., 2014c; Liu et al., 2015; Nguyen et al., 2015). The PBL height
over Wangdu during this campaign was derived from the hybrid single-particle
lagrangian integrated trajectory (HYSPLIT) model (Draxler and Rolph, 2012), which
varied between about 300 m at midnight and over 3000 m at noon. The uncertainty of
our model calculation derives from the uncertainty of observational data. The total
uncertainty in the model was estimated from the errors of all input parameters using
error propagation, which is similar to the method that can be found in Hofzumahaus et
al. (2009). On average, the modelled concentration of atmospheric peroxides had an
uncertainty of approx. 40%. In the present study, to explore the impact of the
heterogeneous process on the concentration of atmospheric peroxides, our box model
is extended with the aerosol uptake of peroxides. The pseudo-first-order rate constant
for the heterogeneous uptake of peroxides on ambient aerosols is parameterized as
follows:
$$k = \frac{1}{4}\gamma \cdot v \cdot S_{aw} \qquad (1)$$



(Jacob, 2000), where $\gamma$ is the uptake coefficient, $v$ is the mean molecular velocity, $S_{aw}$
is the aerosol surface concentration corrected by the measured hygroscopic factor, $f(RH)$
that could be expressed as $S_{aw} = S_a \times f(RH)$.
**3 Results and Discussion**
**3.1 General observations**
The concentrations of peroxides in the atmosphere were measured continuously from 4
June to 7 July 2014. The predominant peroxides over Wangdu included $H_2O_2$, MHP and
PAA. Time series for atmospheric peroxides during Wangdu Campaign 2014 are
illustrated in Fig. 1. The statistical data about the observed concentration of atmospheric
peroxides are summarized and given in Table 1. It should be noted that values below
the detection limit (D.L.) of our instrument were replaced by the corresponding D.L. in
Fig. 1, Fig. 2 and statistical calculations. In this study, $H_2O_2$ accounted for ~70% of
total detected peroxides ($H_2O_2$ + MHP + PAA). However, in our previous work, $H_2O_2$
contributed not more than 30% of total peroxides in the atmosphere over urban Beijing
at the summer time of 2010 and 2011 (Liang et al., 2013b). This might be caused by
the difference on the production and destruction of atmospheric peroxides between two
sites. MHP and PAA were determined to be about 20% and 5% of total peroxides over
Wangdu, respectively, which is similar to the results of other rural sites in China from
our previous investigations (Zhang et al., 2010, 2012).
During this campaign, there were four severe pollution episodes at Wangdu site as
follows: Episode 1 (4 June–6 June), Episode 2 (12 June–17 June), Episode 3 (29 June–
3 July) and Episode 4 (5 July–7 July) with elevated average $PM_{2.5}$ concentrations (75
$\mu g\ m^{-3}$, 92 $\mu g\ m^{-3}$, 79 $\mu g\ m^{-3}$ and 99 $\mu g\ m^{-3}$, respectively). In Episode 1, $H_2O_2$, MHP
and PAA were observed up to 11.3 ppbv, 0.9 ppbv and 1.5 ppbv, respectively. The
maximum $H_2O_2$ concentration on 5 June was the highest value so far among the
previously reported observations in urban, suburban and rural areas in China at summer
time. The possible reason for this peak concentration at Wangdu site could be the
primary emission from biomass burning combined with the secondary formation by the
intense photochemical process. Nevertheless, owing to the lack of supporting data for





other pollutants and parameters, it is difficult to identify the relative contributions of
biomass burning versus photochemical formation to the burst of atmospheric peroxides
on 5 June. In Episode 2, there was widespread and intensive biomass burning in the
NCP as this observation period covered the local wheat harvest season. The sudden
raise of atmospheric peroxides was observed and further discussed in Sect. 3.3. In
Episode 3, there was a substantial decline of $H_2O_2$, MHP and PAA level during this
typical haze event compared with the foregoing two episodes, which can be ascribed to
the heterogeneous uptake of peroxides on atmospheric aerosols on haze days over
Wangdu (See Sect. 3.4). In Episode 4, Wangdu was significantly impacted by the
regional transport (Ye, 2015). The concentrations of atmospheric peroxides remained
relatively low compared with the other three episodes. In addition to the above-
mentioned episodes, it was relatively clear between 8 June and 11 June and 27 June and
28 June, with mean $PM_{2.5}$ concentrations under 40 μg m$^{-3}$. The intermittent
thunderstorm activities occurred from 19 June to 25 June that caused the electric power
failure and several data gaps.

**3.2 Peroxide simulation**

In this study, we employed a box model based on the MCMv3.3.1 to simulate $H_2O_2$,
MHP and PAA concentrations. Here, to explore the atmospheric chemistry of peroxides
on non-haze, biomass burning and haze days, the observational data from 8 June to 11
June (Phase I), from 15 June to 17 June (Phase II) and from 29 June to 3 July (Phase
III) in 2014 were selected as phase of interest and analyzed in detail using box model
in the following sections. The temporal variations of meteorological parameters,
chemical species and atmospheric peroxides for these three phases are displayed in Fig.
2. The observed and calculated levels of atmospheric peroxides for the three phases are
illustrated in Fig. 3. During these case study phases, 75% of the wind speed data were
≤ 2.2 m s$^{-1}$ and the mean value was 1.6 m s$^{-1}$. It has been shown that the atmospheric
lifetimes of peroxides are on the order of several hours as reported previously (He et
al., 2010; Wu et al., 2015), implying that the effect of regional transport or dilution on
the concentrations of atmospheric peroxides was of little significance over Wangdu.




Hence, the regional-scale transport can be excluded in our box model and the budgets
of peroxides are, to a large extent, dependent on local chemical processes during the
observation. In the Phase I, as shown in Fig. 4, the model prediction had good
performance in the daytime (06:00–18:00 local time), which was 1–2 times higher than
the measurement results. This seems to be explained by the model-measurement
uncertainty. Similarly, a previous observation carried out at a suburban site also showed
reasonable model-measurement agreement in $H_2O_2$ level on sunny days (Guo et al.,
2014). The excellent description yielded by the model base case indicated that the
production and destruction of atmospheric peroxides on non-haze days were calculated
correctly based on the current understanding of atmospheric peroxide related chemistry.
However, the simulation in the nighttime (18:00–06:00 local time) during the Phase I
demonstrated an obvious overestimation compared to the observation by a factor of 4–
6 and up to an order of magnitude. This large discrepancy between calculated and
observed results is speculated to be resulted from the underestimation of sink terms as
the key precursors governing the formation of atmospheric peroxides are constrained
by the observation and the overestimation of source terms can be ruled out by assuming
that the chemical mechanisms of atmospheric peroxides are well-understood. It
coincides with the comparison of the simulated and observed $H_2O_2$ concentration over
urban Beijing, in which the explanation for the overprediction of $H_2O_2$ level on haze
days was thought to be the heterogeneous processes on liquid or solid particles that
were missing from the current atmospheric chemistry model (Liang et al., 2013b).
Considering the high aerosol loading in the NCP and the higher aerosol surface area
concentration at nighttime than that at daytime in the Phase I, we believe that the
missing sink for atmospheric peroxides in the model base case is probably
heterogeneous uptake of peroxides occurring on aerosols. The strength of the missing
sink for $H_2O_2$, MHP and PAA were estimated to be 0.24, 0.09, 0.03 ppbv h$^{-1}$ on average,
respectively, which was on the same order of magnitude as the known loss rates of
atmospheric peroxides during the Phase I. In the Phase II, the comparison of the
modelled and measured peroxide concentrations in Fig. 3 displays that the observed
magnitude of atmospheric peroxides was unexpectedly large, indicating a missing



source for peroxides. Such a strong imbalance was found only in the Phase II during
the whole campaign, and the measurement-to-model ratio based on the model case was
up to a factor of 7 for MHP on 17 June, which was much higher than the measurement
and model errors. In the past, the higher-than-expected concentrations of atmospheric
peroxides have also been reported by Lee et al. (1997), in which $H_2O_2$, MHP, PAA and
other organic peroxides levels elevated near biomass burning plumes. Given the
frequent fire emissions in the NCP during the Phase II that are quite similar to the
conditions in Lee et al. (1997), it appears that the significant mismatch can be attributed
to the direct production from biomass burning (See Sect. 3.3). In the Phase III, the
calculated values in the model base case showed a general tendency to strongly
overestimate the observed values (Fig. 3). As there was a typical haze event during the
Phase III, the model-measurement imbalance was probably due to the missing sink for
atmospheric peroxides, which was the same deficiency in the model as that in the Phase
I. It can be seen in Fig. 3 that with the inclusion of heterogeneous reactions on aerosol
particles, the simulated concentrations of atmospheric peroxides were apparently
improved, which is further quantified in Sect. 3.4.

299       Before exploring the impact of biomass burning and heterogeneous uptake on the

chemistry of atmospheric peroxides, we performed a model test by implementing the
newly proposed chemical mechanisms for $CH_3C(O)O_2$ and $CH_3O_2$ related chemistry in
MCMv3.3.1, as listed in Table 2. The rate constant and the branching ratios of the
$CH_3C(O)O_2$ + $HO_2$ reaction that was the major pathway for the formation of PAA in
this model scenario were modified according to the recent laboratory study conducted
by Winiberg et al. (2016). Additionally, we also incorporated the reaction between
$CH_3O_2$ radicals and OH radicals that has as yet seldom been involved in atmospheric
chemistry model because it was recognized as an important sink for $CH_3O_2$ radicals
with non-negligible effect on subsequent formation of MHP under remote conditions
by Bossolasco et al. (2014) and Fittschen et al. (2014), in spite of the fact that the
reaction product is still unknown. As shown in Fig.3, the model run containing newly-
proposed mechanisms did not have a remarkable influence on the simulated results of
$H_2O_2$ in comparison to the model base case. But a slight difference of up to ~20%





between calculated and observed MHP can be noted at night, resulting from the
additional removal pathway of $CH_3O_2$ radicals from the noon to the sunset. The increase
of over 70% in rate constant and the reduction of about 10% in the branching ratio of
the reaction $CH_3C(O)O_2 + HO_2 \rightarrow CH_3C(O)OOH$ generated systematically 1.5 times
higher PAA concentration in this model scenario than that in the model base case.
Nevertheless, although the modelled PAA during the Phase II can be raised close to the
level of the observation, the concentrations of atmospheric peroxides were not fully
captured by the model with the implementation of newly proposed mechanisms (Fig.
3). Moreover, the resulting MHP and PAA values still agreed with the measurements
in the range of their errors. Thus, we can conclude that the additional chemical
mechanisms embedded in the model only have a marginal impact that is not sufficient
to match the observed peroxides in the atmosphere. The efficient source or sink for the
reproduction of the observation will be deeply investigated below.
As outlined in the introduction, the source of $H_2O_2$, MHP and PAA are the direct
emission from biomass burning and the photochemical oxidation of VOC precursors
via $HO_2$, $CH_3O_2$ and $CH_3C(O)O_2$ formation. However, it is still difficult to determine
the contributions of VOC precursors at a species level. Here, to gain further insight into
the secondary chemical transformation of atmospheric peroxides at Wangdu site, the
sensitivity study with an indirect approach adopted referring to the relative incremental
reactivity (RIR) concept for ozone formation in Cardelino and Chameides (1995) was
utilized to track out the major VOC precursors of atmospheric peroxides and assess
their roles by the numerical model with the application of the MCMv3.3.1 that can
describe the explicit degradations of VOC species and quantify their contributions
individually. In this work, the definition of RIR is the ratio of reduction in the
production rates of atmospheric peroxides to the reduction of VOC precursor
abundances by 25% compared to the model base case, which can be regarded as a proxy
for the influence of a specific VOC on the *in-situ* formation of atmospheric peroxides.
The Phase I and Phase III were selected for the analysis, while the Phase II was
precluded from the analysis as it was affected by the local emission that was disregarded
in the model base case. Fig. 5 displays the average RIRs of $H_2O_2$, MHP and PAA for



alkane, alkene, aromatic and $NO_x$ classes as well as seven most important individual
VOC precursors. The results demonstrate that the formation of $H_2O_2$ was sensitive to
alkenes and insensitive to alkanes, aromatics and $NO_x$. The production of MHP and
PAA shows strong dependence of alkenes and $NO_x$, while it is relatively independent
of aromatics and alkanes other than methane. In terms of VOC species with relatively
high RIR that is more than 0.001 for $H_2O_2$ as well as more than 0.01 for MHP and PAA,
it is seen that isoprene from the local biogenic emission and trans-2-butene from the
anthropogenic emission turn out to be the key VOC species controlling the formation
of atmospheric peroxides. Besides, cis-2-butene, cis-2-pentene, propene and 1,2,4-
trimethylbenzene also seem to be the major individual VOC precursors as evidence by
Fig. 5. Methane is noticed to be an important contributor to the formation of MHP. Such
list of VOC species is not consistent with our previous studies over urban Beijing that
suggested aromatics (i.e., toluene and dialkylbenzenes) as the dominant VOC precursor
of atmospheric peroxides (Zhang et al., 2010; Liang et al., 2013b). It reflects that the
relative significance of individual VOC precursors varies from place to place. The
distinction between two sites is attributable to the relatively more abundant isoprene,
anthropogenic alkenes and much less reactive aromatics at the rural site in the NCP
than those at the urban site, Beijing. With the support on the basis of the identification
of a small class of key VOC precursors contributing to the formation of peroxides in
the atmosphere of NCP, the effective control strategies for mitigating the pollution
resulted from atmospheric peroxides can be formulated. In the NCP, it has been
revealed that the vehicular exhaust is the predominant source for the responsible VOC
species such as propene, trans/cis-2-butenes and trimethylbenzenes in the surrounding
areas of the observation site (Yuan et al., 2009; Ran et al., 2011; Li et al., 2014b; Li et
al., 2015; Wu et al., 2016), while the vegetation governs the release of isoprene. Taking
into account the fact that biogenic emissions of isoprene are not controllable, it is
recommended to take measures for vehicle emission reduction in order to mitigate the
pollution of atmospheric peroxides in the NCP and hence alleviate their potential
harmful effects on air quality, human health and ecosystem.



### 3.3 Direct production of peroxides from biomass burning

In the Phase II, the levels of $H_2O_2$, MHP and PAA were highly elevated in comparison

with the other phases, which could not be explained by the photochemical process in

the model base case alone. It provides us a hint that an additional formation pathway is

required to improve the results of model simulation. In Sect. 3.2, we hypothesized that

the direct production of peroxides from biomass burning should serve as an essential

source for the unexpected burst of atmospheric peroxides. Here, we tested the

hypothesis by means of the box model and linear regression with the observation data

from three events mentioned below during the Phase II. It is well known that CO and

$K^+$ can be used as the reference for the biomass combustion (Koppmann et al., 2005;

Reid et al., 2005; Li et al., 2007; Sullivan et al., 2008; Cheng et al., 2013, 2014; Li et

al., 2014a; Wang et al., 2015). The averaged CO levels were 0.42±0.16 ppmv,

0.79±0.20 ppmv and 0.61±0.20 ppmv for the Phase I, Phase II and Phase III,

respectively. The mean $K^+$ concentrations were about 0.64±1.19 µg m$^{-3}$ for the Phase

I, 2.51±1.53 µg m$^{-3}$ for the Phase II and 0.26±0.21 µg m$^{-3}$ for the Phase III. The

abundance of CO and $K^+$ during the Phase II increased apparently compared with that

during the Phase I and Phase III, which was consistent with the observed intensive

biomass burning activities at Wangdu site (Ye, 2015). Nevertheless, in addition to the

biomass burning, CO level in the NCP was also affected by anthropogenic activities

with the regional transport of polluted air masses, for example, the urban plumes. It has

been proved that airborne $K^+$ is acceptable as the tracer for biomass burning during

summertime in the NCP (Cheng et al., 2013; Wang et al., 2015). Therefore, $K^+$ might

be a better indicator of biomass burning than CO here. In the Phase II, we identified

several biomass burning events with the concentration of $K^+$ twice more than the mean

value of that in the Phase I and Phase III. Considering the availability of the observation

data for atmospheric peroxides, we focused our analysis on three events as follows:

Event I (17:00–20:00 on 15 June), Event II (22:00 on 16 June–1:00 on 17 June) and

Event III (12:00–15:00 on 17 June) with the duration of over 3 hours.

As illustrated in Fig. 3, the model base case cannot reproduce the measurements for



atmospheric peroxides in the three events. To match the observation, the primary
sources for $H_2O_2$, MHP and PAA were applied to our model. The strengths of the
primary sources were calculated to be about 0.25–0.98 ppbv h$^{-1}$, 0.09–0.44 ppbv h$^{-1}$
and 0.02–0.14 ppbv h$^{-1}$ for $H_2O_2$, MHP and PAA, respectively. These values were on
the order of the known secondary production rates of atmospheric peroxides during the
three events. It should be pointed out that the estimation was associated with large
uncertainties since it did not include the heterogeneous uptake of peroxides by aerosols
in the model here. In view of the possible additional sink for atmospheric peroxides as
discussed in Sect. 3.4 below, the primary sources for $H_2O_2$, MHP and PAA might
represent the lower limit. The effect of biomass burning on the levels of atmospheric
peroxides might be underestimated as well. We underscore that there might exist even
larger missing sources for $H_2O_2$, MHP and PAA due to the scarcity of some important
removal pathways of atmospheric peroxides in the model in this section.
The results of linear regression involving correlation coefficients and their statistical
significance of $H_2O_2$, MHP and PAA to CO and $K^+$ were listed in Table 3 for the three
biomass burning events. The relationships between atmospheric peroxides and biomass
burning indicators were analyzed separately for each event owing to the variability of
fire emissions. A notable trend between atmospheric peroxides and $K^+$ was found with
correlation coefficients exceeding over the significance threshold, which provided a
convincing evidence for the direct production of peroxides from biomass burning as the
additional source. Moreover, it was noticed that CO coincided well with $K^+$ for the
Event I and Event II, exhibiting excellent correlation with atmospheric peroxides (Table
3). The enhancement ratios relative of $H_2O_2$, MHP and PAA to CO were calculated to
be at the magnitude ranging from $10^{-3}$ to $10^{-2}$, which were similar to the enhancement
signals of atmospheric peroxides to CO obtained near biomass fires from flights
published by Lee et al. (1997).
It is noteworthy that several other chemical processes, for example, secondary
formation via the photooxidation of potential unmeasured short-lived VOC species
emitted from biomass fires prior to our sampling of the plume at the observational site
seem to be the alternatives to the direct production from biomass burning as the missing





source of atmospheric peroxides in the model. Thus, it appears necessary and desirable
to further distinguish the extent to which atmospheric peroxides are generated via the
direct production or secondary formation from biomass burning in future research.
Laboratory studies are required to simulate the biomass fires in the NCP using
combustion chamber to critically characterize the emission factors of atmospheric
peroxides to CO and determine their generation mechanisms. Also, more reliable
aircraft and ground-based field measurements for the variation of atmospheric
peroxides during the harvest seasons in China need to be carried out and would be
beneficial to shed some light on the role of biomass burning in the abundance of
peroxides in the atmosphere.
**3.4 Heterogeneous uptake of peroxides by aerosol**
In Sect. 3.2, heterogeneous uptake on atmospheric particles was considered as a suitable
explanation for the missing sink for $H_2O_2$, MHP and PAA during the Phase I and Phase
III in view of substantial aerosol loading in the NCP that provided considerably sites
for heterogeneous reactions. Here, we made an attempt to implement a parameterization
of heterogeneous uptake by aerosols in our box model to resolve the deviation between
the simulated and observed data (See Sect. 2.3). Using the uptake coefficient of $1 \times 10^{-3}$
for $H_2O_2$, MHP and PAA, a good agreement between the modelled and measured
temporal variation of atmospheric peroxides can be obtained in Phase I and Phase III
by taking into account the combined model-measurement error that is conservatively
assumed to be ~50% (Fig. 3). The calculated $H_2O_2$, MHP and PAA with the coupling
of heterogeneous reaction was on average decreased by about 75% compared to the
results in the model base case during the Phase III. The uptake coefficient of $1 \times 10^{-3}$
approached the upper limit of the laboratory measured value for $H_2O_2$ on mineral dust
($9 \times 10^{-4}$) reported by Pradhan et al. (2010), but a little higher than the previous
measured values on ambient $PM_{2.5}$ of $(1-5) \times 10^{-4}$ during the summertime over urban
Beijing (Wu et al., 2015). It is reasonable as Wu et al. (2015) pointed out that the uptake
coefficients for $H_2O_2$ and organic peroxides on ambient $PM_{2.5}$ are in the same range
and show no obvious differences between daytime and nighttime or between non-hazy





and hazy conditions.
With the adoption of heterogeneous uptake coefficients of $1 \times 10^{-3}$, we evaluated the
sinks of atmospheric peroxides in the Phase I and Phase III that represented non-haze
and haze conditions, respectively. The mean surface area concentration that was
corrected for the hygroscopic growth of aerosol was measured to be 968 $\mu m^2\ cm^{-3}$ for
Phase I and 1491 $\mu m^2\ cm^{-3}$ for Phase III. Fig. 6 demonstrated that the destruction of
atmospheric peroxides during the two phases originated from a diversity of sinks,
including photolysis, OH-initiated reaction, dry deposition and heterogeneous uptake.
It has been reported that heterogeneous reaction is the most important sink for $H_2O_2$ in
urban (Liang et al., 2013b) and suburban areas (Guo et al., 2014). In contrast, OH-
initiated reaction and dry deposition were regarded as the major removal pathways of
organic peroxides in rural (Zhang et al., 2012) and forests areas (Nguyen et al., 2015).
Here, heterogeneous uptake by aerosols turned out to be the predominant sink for
atmospheric peroxides in the NCP, accounting for more than 60% of the total loss,
while dry deposition became the marginal removal pathway that contributed ~10% to
the destruction of $H_2O_2$, MHP and PAA. The role of OH-initiated reaction in the total
loss varied between the speciated peroxides with no more than 30%. Photolysis only
represented a minor contribution (<3%). The most prominent feature on haze days was
the larger loss of atmospheric peroxides via heterogeneous process, demonstrating the
enhanced impact of aerosols on the sink of peroxides during the haze episode compared
to that during the non-haze episode. On the basis of the analysis above, we investigated
the atmospheric lifetime of peroxides in the NCP with the integration of observation
and modelling. The lifetime of $H_2O_2$, MHP and PAA were estimated with the
concentration-to-time curves between 18:00 and 24:00 LT as the formation of
atmospheric peroxides was weak and negligible during this phase. The average lifetime
obtained from the field observation between 18:00 and 24:00 LT in the Phase I was
around 4.0 h, 5.6 h and 3.1 h for $H_2O_2$, MHP and PAA, respectively, which was similar
to the values of 3.4 h, 4.3 h and 5.2 h for $H_2O_2$, MHP and PAA, respectively, given by
our modeling simulation. The lifetime of atmospheric peroxides in the Phase III was



~40% smaller than that in the Phase I. Using the box model, the atmospheric lifetime
of $H_2O_2$, MHP and PAA during the whole Phase I and Phase III was calculated to be
about 2.1 h, 2.3 h and 3.0 h, respectively. This is comparable to the literature results
with the inclusion of heterogeneous reaction (Liang et al., 2013b; Wu et al., 2015), but
notably shorter than the recent studies conducted by Khan et al. (2015) and Nguyen et
al. (2015) without the coupling of the heterogeneous process. The simulated lifetime of
atmospheric peroxides can be over 10 h by supposing that the loss of $H_2O_2$, MHP and
PAA is merely due to photolysis, OH-initiated reaction and dry deposition. It
emphasizes that heterogeneous uptake on aerosols determines the atmospheric lifetime
of peroxides.
It is worth noting that the heterogeneous uptake of peroxides by aerosols in the
atmospheric chemical model is still controversial as it is possibly that the aerosol uptake
of $HO_2$ radicals is the explanation for the missing sink. This raises an interesting
question of whether $HO_2$ uptake or peroxide uptake is responsible for the imbalance
between observation and modelling. It has been referred by formerly published
literature that aerosol uptake of $HO_2$ radicals is the major reason for the overprediction
of the levels of atmospheric peroxides in the model (de Reus et al., 2005; Mao et al.,
2013; Guo et al., 2014). Nevertheless, it is apparent that the extent of $HO_2$
heterogeneous degradation depends on the atmospheric environment, especially the
concentration and property of aerosol particles that are various under different
conditions. The measured and modelled $HO_2$ concentrations at Wangdu site are close
to each other, implying that the budget of $HO_2$ is well captured by the box model merely
with the gas-phase regional atmospheric chemical mechanism (RACM) comprised (K.
Lu, personal communication, 2015). Hence, aerosol uptake of $HO_2$ radicals is
insignificant during Wangdu Campaign 2014 and not taken into account in our model,
while heterogeneous uptake of atmospheric peroxides by aerosols is exclusively
adopted to improve the reproduction of the observation in the two phases above. It has
been inferred that heterogeneous uptake of peroxides on ambient $PM_{2.5}$ is probably
resulted from solid surface reactions and aerosol aqueous reactions (Wu et al., 2015),



for instance, "Fenton-like" reaction between peroxides and transition metal ions, which
is supported by the laboratory studies (Chevallier et al., 2004; Deguillaume et al., 2005)
and field observation (Liang et al., 2013b; Guo et al., 2014). Nevertheless, the detailed
heterogeneous mechanism containing individual reaction channels was not included in
the present work owing to the chemical complexity of the ambient aerosol. Given the
potential importance of atmospheric peroxide compounds on the generation of $HO_x$
radicals and aerosol ROS, the aging of mineral dust and SOA and the formation of haze
(Huang et al., 2015; Pöschl and Shiraiwa, 2015; Zhang et al., 2015; Li et al., 2016),
more comprehensive investigations including laboratory, field and modelling studies
on the heterogeneous uptake processes of $H_2O_2$, MHP, PAA and other peroxides are
indispensable to provide concrete evidence to elucidate the chemical budget of
atmospheric peroxides in the future.
**4 Conclusions**
Atmospheric peroxides including $H_2O_2$, MHP and PAA were measured at a rural site
during Wangdu Campaign 2014. The maximum $H_2O_2$ concentration was observed to
be 11.3 ppbv, which was the highest value compared with previous observations in
China. The concentrations of atmospheric peroxides were highly elevated during the
biomass burning activities, but underwent substantial decline during the haze events.
With the application of observation-based model combining measured meteorological
parameters and trace gases, we analyzed the chemical budget of peroxides under
biomass burning, non-haze and haze conditions. Photochemical formation of
atmospheric peroxides was attributed to a small class of alkenes, while it was
insensitive to alkanes and aromatics. The key VOC precursors controlling the formation
of peroxide compounds were identified to be isoprene, trans/cis-2-butenes, cis-2-
pentene, propene and trimethylbenzene. The base model simulation (MCMv3.3.1)
underpredicted the levels of atmospheric peroxides up to a factor of 7 during biomass
burning events compared with the measurement. The direct production from biomass
burning was regarded as the explanation for the unexpected burst of peroxides. To
improve the simulated concentrations, the strengths of the primary emissions from



biomass burning should be on the same order of the known secondary production rates
of atmospheric peroxides. Moreover, the model base case also overpredicted the
concentrations of atmospheric peroxides on haze days in comparison with the
observation. The heterogeneous uptake by aerosols was suggested to be responsible for
the attenuation of peroxides. The model could reproduce the observed values with the
introduction of heterogeneous process using the uptake coefficient of $1 \times 10^{-3}$ for
atmospheric peroxides. According to the closure between observed and calculated
concentrations, the heterogeneous uptake on aerosol particles was found to be the
predominant sink for atmospheric peroxides, accounting for more than 60% of the total
loss, followed by the OH-initiated reaction (<30%) and dry deposition (~10%). The
mean atmospheric lifetime of peroxides in summer in the NCP was estimated to be
around several hours that was in good agreement with previous laboratory studies for
the aerosol uptake of peroxides, indicating that heterogeneous reaction determines the
atmospheric lifetime of peroxides. In view of the importance of peroxides in
tropospheric oxidation capacity and formation potential of secondary aerosols, more
reliable investigations focused on the biomass burning emission factors and detailed
heterogeneous mechanism of speciated peroxides are urgently required to further
quantitatively evaluate the role of biomass burning and heterogeneous uptake in the
abundance as well as budget of atmospheric peroxides and facilitate our knowledge of
the formation of haze pollution.
*Acknowledgements.* This work was funded by the National Natural Science Foundation
of China (grants 41275125, 21190051, 21190053, 21477002, and 41421064). The
authors would like to thank Min Shao group (Peking University) for their VOCs data
and Alfred Wiedensohler group (Leibniz Institute for Tropospheric Research) for their
particle surface area concentrations data. The authors wish to gratefully thank the entire
Wangdu Campaign 2014 team for the support and collaboration at Wangdu site.

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



**Table 1.** Summary of the concentrations of atmospheric peroxides during Wangdu Campaign 2014.

|  |  | $H_2O_2$ (ppbv) | MHP (ppbv) | PAA (ppbv) |
|---|---|---|---|---|
|  | D.L.[a] | 0.01 | 0.01 | 0.01 |
| 24 h | N[b] | 1797 | 1797 | 1797 |
|  | Mean | 0.51 | 0.17 | 0.04 |
|  | S.D.[c] | 0.90 | 0.20 | 0.11 |
|  | Median | 0.19 | 0.11 | 0.01 |
|  | Maximum | 11.3 | 1.25 | 1.49 |
| Daytime (06:00–18:00 LT[d]) | N[b] | 829 | 829 | 829 |
|  | Mean | 0.55 | 0.16 | 0.03 |
|  | S.D.[c] | 0.83 | 0.17 | 0.12 |
|  | Median | 0.24 | 0.12 | 0.01 |
|  | Maximum | 10.20 | 1.20 | 1.49 |
| Nighttime (18:00–06:00 LT[d]) | N[b] | 968 | 968 | 968 |
|  | Mean | 0.48 | 0.17 | 0.04 |
|  | S.D.[c] | 0.96 | 0.23 | 0.11 |
|  | Median | 0.15 | 0.11 | 0.01 |
|  | Maximum | 11.33 | 1.25 | 1.47 |

[a] D.L.: detection limit.

[b] N: number of samples.

[c] S.D.: standard deviation.

[d] LT: local time.



**Table 2.** Chemical mechanisms for $CH_3C(O)O_2$ and $CH_3O_2$ related chemistry modified

or added to MCMv3.3.1.

| Reactions | Rate constants ($cm^3$ molecule$^{-1}$ s$^{-1}$) | Reference |
|---|---|---|
| **$CH_3C(O)O_2$ chemistry** | | |
| $CH_3C(O)O_2 + HO_2 \rightarrow CH_3C(O)OOH + O_2$ | $2.40 \times 10^{-11} \times 0.37$ | Winiberg et al. (2016) |
| $CH_3C(O)O_2 + HO_2 \rightarrow CH_3C(O)OH + O_3$ | $2.40 \times 10^{-11} \times 0.12$ | Winiberg et al. (2016) |
| $CH_3C(O)O_2 + HO_2 \rightarrow CH_3 + CO_2 + OH + O_2$ | $2.40 \times 10^{-11} \times 0.51$ | Winiberg et al. (2016) |
| **$CH_3O_2$ chemistry** | | |
| $CH_3O_2 + OH \rightarrow PRODUCT$ | $2.80 \times 10^{-10}$ | Fittschen et al. (2014) |





**Table 3.** Linear regression of atmospheric peroxide species to CO and $K^+$ for three biomass burning events during the Phase II (15 June–17 June). Correlation coefficients shown in italic and bold indicate statistical significance (p<0.05) and higher statistical significance (p<0.01), respectively.

| Species | Slope [a] | Correlation coefficient | | $N^{[b]}$ | Critical correlation coefficient |
|---|---|---|---|---|---|
| | | CO | $K^+$ | | |
| *Event I* | | | | | |
| $H_2O_2$ | $2.17 \times 10^{-3}$ | **0.8144** | **0.8432** | 10 | 0.7646 ($p < 0.01$), |
| MHP | $1.23 \times 10^{-3}$ | *0.6873* | *0.7624* | 10 | 0.6319 ($p < 0.05$) |
| PAA | $7.16 \times 10^{-4}$ | **0.8378** | **0.9515** | 10 | |
| *Event II* | | | | | |
| $H_2O_2$ | $1.32 \times 10^{-2}$ | **0.9134** | **0.8538** | 11 | 0.7348 ($p < 0.01$), |
| MHP | $2.30 \times 10^{-3}$ | **0.8876** | *0.7042* | 11 | 0.6021 ($p < 0.05$) |
| PAA | $6.73 \times 10^{-4}$ | **0.8399** | 0.5330 | 11 | |
| *Event III* | | | | | |
| $H_2O_2$ | N/A [c] | N/A [c] | **0.9632** | 9 | 0.7977 ($p < 0.01$), |
| MHP | N/A [c] | N/A [c] | **0.8741** | 9 | 0.6664 ($p < 0.05$) |
| PAA | N/A [c] | N/A [c] | **0.8436** | 9 | |

[a] Slope: enhancement ratio of speciated peroxides relative to CO.

[b] N: number of samples.

[c] N/A: missing data.





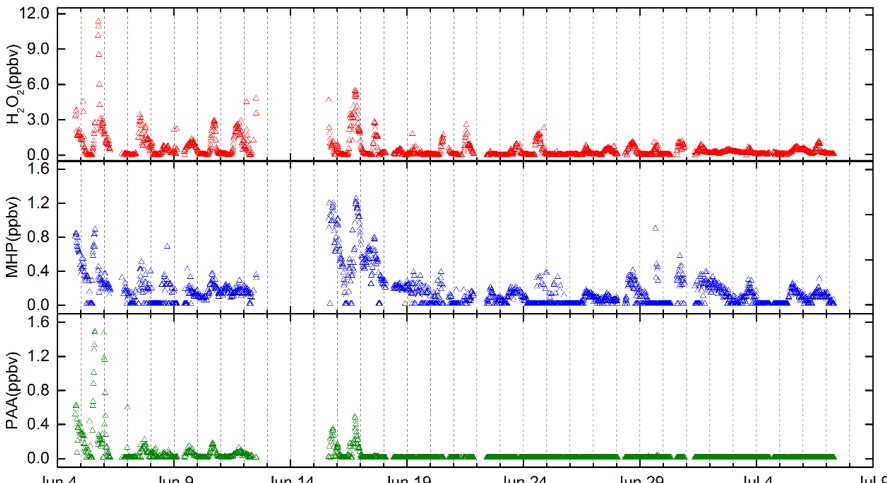

**Figure 1.** Temporal profile for atmospheric peroxides over the entire Wangdu Campaign 2014.



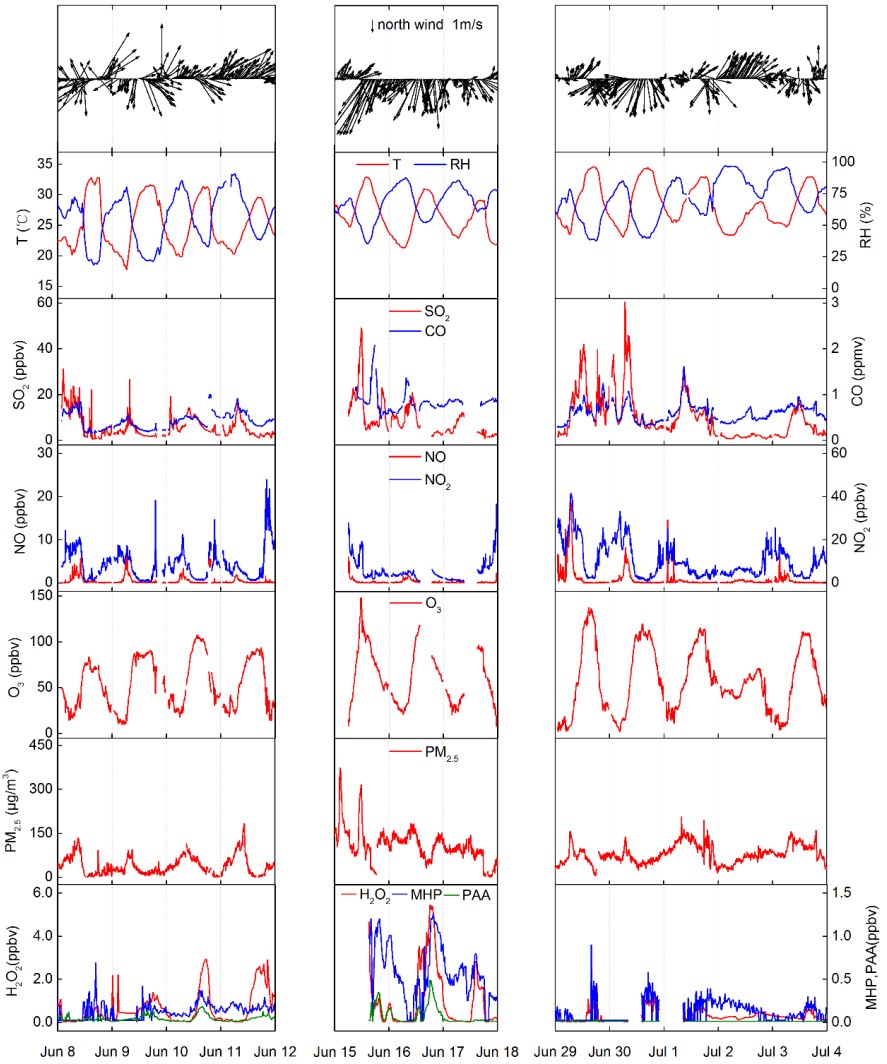

**Figure 2.** Time series of meteorological parameters, chemical species and atmospheric peroxides for Phase I (8 June–11 June), Phase II (15 June–17 June) and Phase III (29 June–3 July).





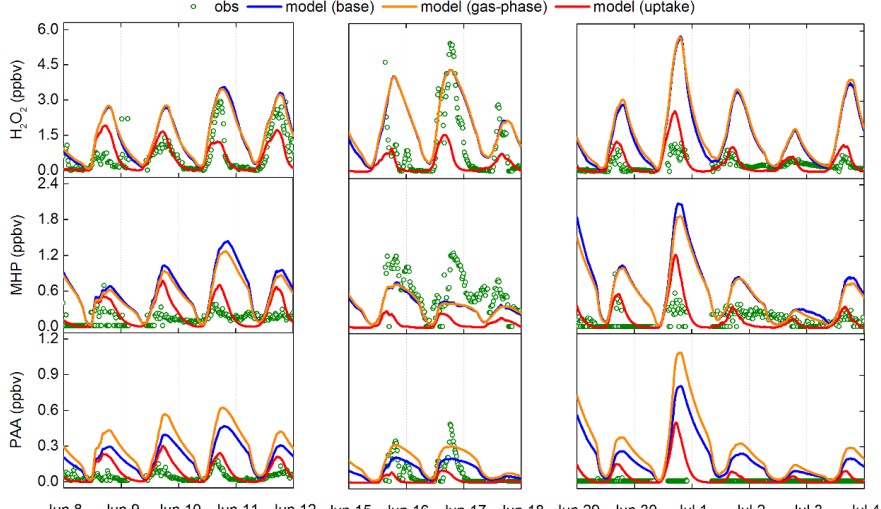

**Figure 3.** Observed and modelled concentrations of atmospheric peroxides for Phase I (8 June–11 June), Phase II (15 June–17 June) and Phase III (29 June–3 July). The green circles represent observed concentrations. The blue, orange and red lines indicate the modelled concentrations from three different scenarios: base case, new gas-phase reaction case and heterogeneous uptake case, respectively.





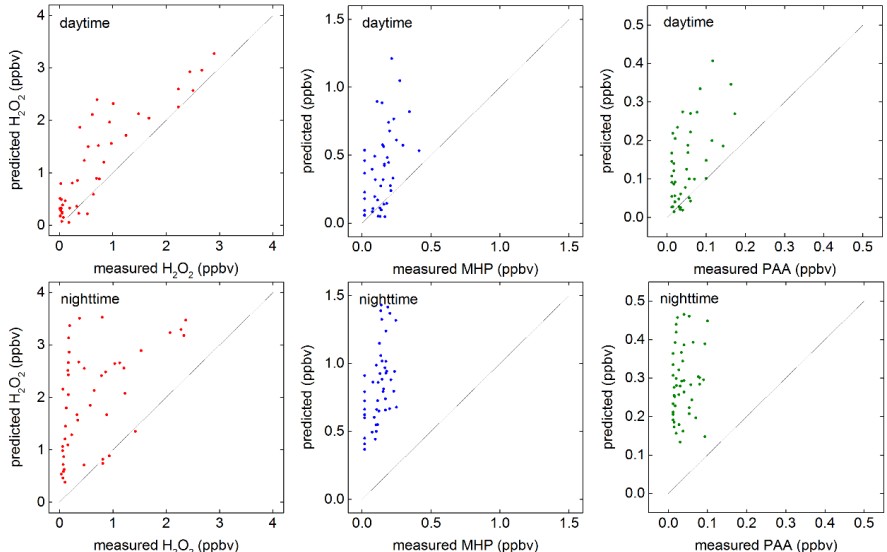

**Figure 4.** Comparisons between measured and predicted concentrations of atmospheric peroxides for daytime and nighttime during the Phase I (8 June–11 June). The solid lines represent the 1:1 ratio of observed to modelled values.



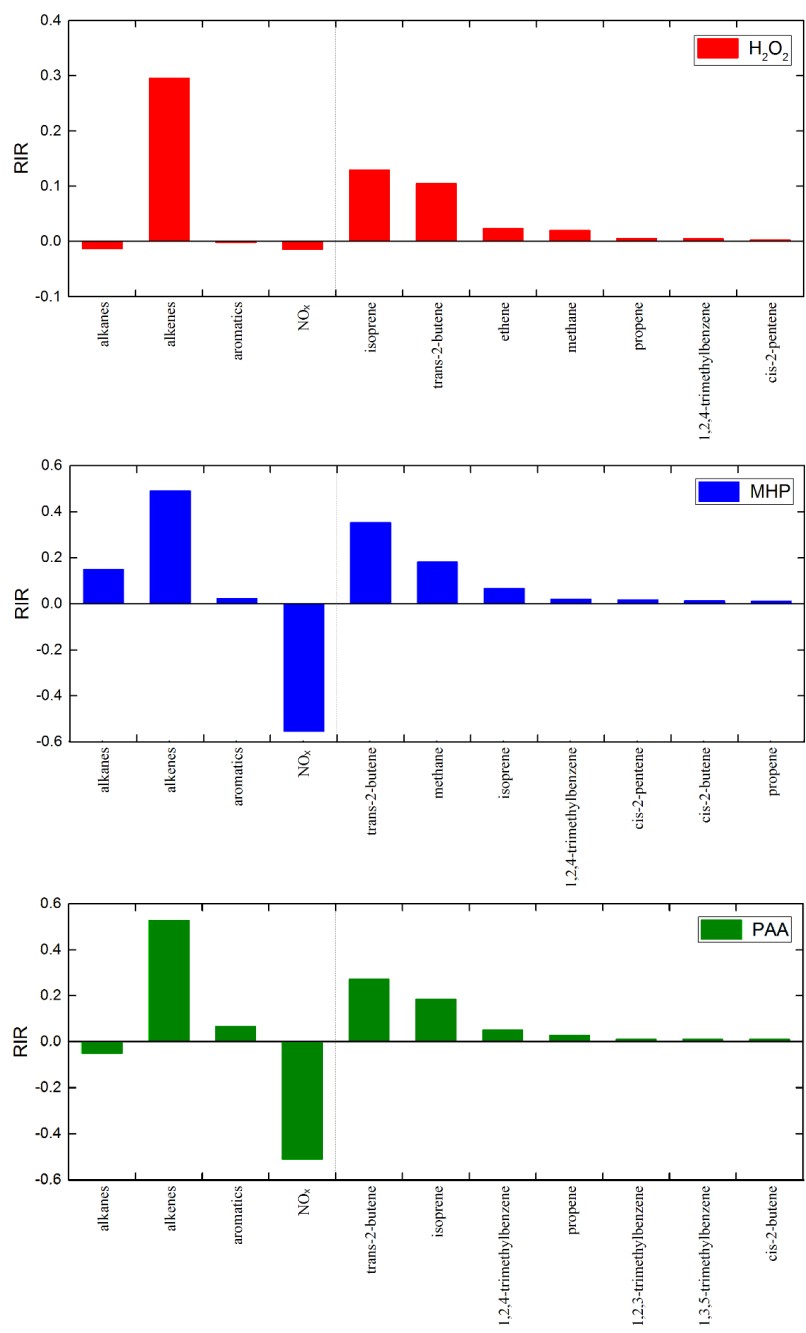

**Figure 5.** Sensitivity of production rate of atmospheric peroxides to major VOC precursor groups and individual VOC species for Phase I and Phase III.




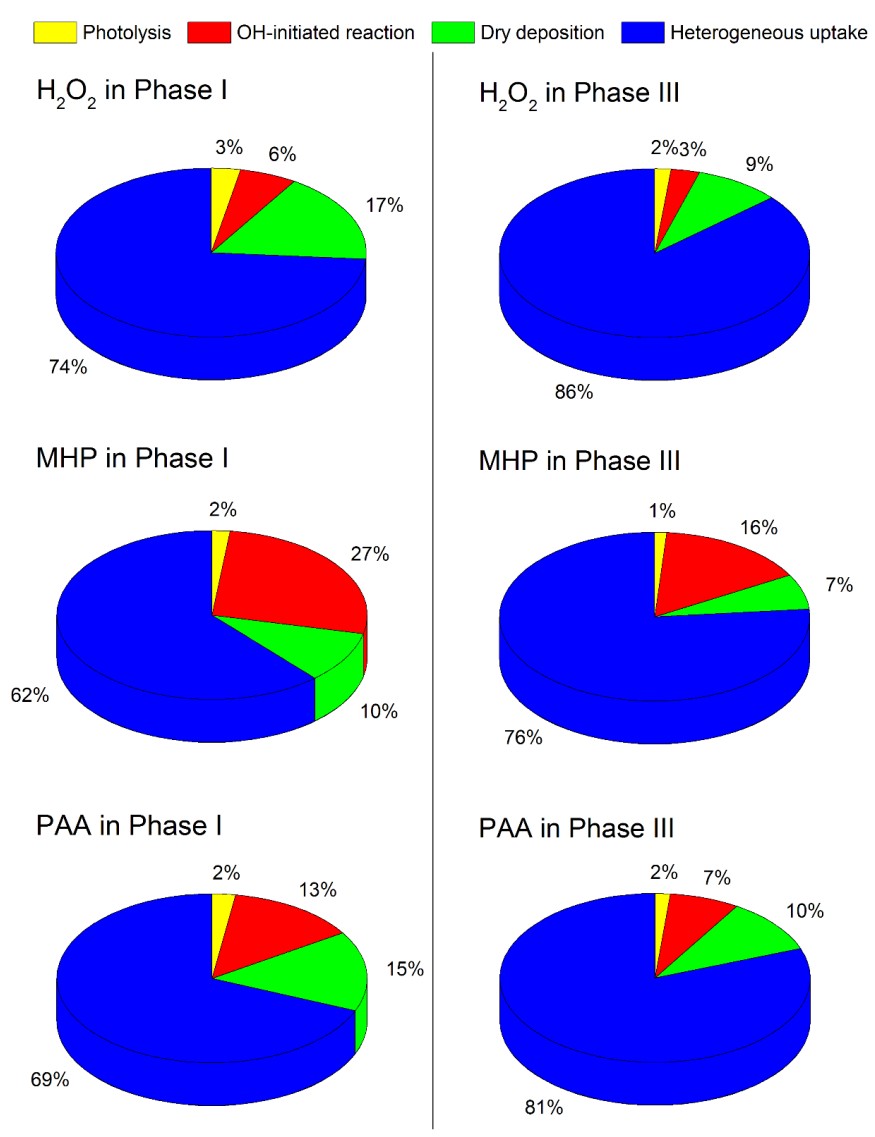

**Figure 6.** Contributions of each sink to $H_2O_2$, MHP and PAA destruction in the box model with the heterogeneous uptake by aerosols added during Phase I and Phase III.