# Peer review of "Observation of atmospheric peroxides during Wangdu Campaign"

_Atmospheric Chemistry and Physics, 2016_

## Referee Comment (RC1) · Anonymous Referee #1 · 1 Jun 2016

Overall Comments

This paper presents some interesting ideas and potentially some interesting evidence for direct emissions of peroxides from biomass burning and heterogeneous loss of peroxides. However, it is far from clear how the model was set up, nor how several of the calculated values were derived. This leaves many open questions when it comes to the data analysis. The interpretation appears to focus on some features of the data and ignoring others, without clear justification. Showing more of the data would better justify the selection of the different phases and episodes and would better illustrate how they differed from each other. This would then strengthen the scientific findings.

I have therefore recommended major changes, as I think there are a number of important changes that are required to bring the paper up to a scientifically robust standard to be published in ACP.

Specific Comments

In the model description it seems to suggest that the uncertainty in the model is just due to the uncertainty in the observational constraints (linee184-5). What about uncertainties in chemical mechanism - rate coefficients, reaction products, reaction simplifications?

Dry deposition is a major sink for peroxides, especially H2O2. What is the uncertainty in the PBL height? Is this included in the total model error?

Assuming that data below the D.L. is equal to the D.L will lead to an overestimation on average (lines 204-5). I would suggest using a value of 0.5 D.L.

Lines 211-13: If MHP and PAA being about 20% and 5% of the total peroxides in Wangdu is similar to the results from other rural sites in China, then presumably it could also be stated that with 70% of the peroxides in Wangdu being H2O2 that Wangdu is also similar to other rural sites in China?

Lines 214-245: It would be helpful to provide more clarity on what is meant by severe pollution episodes (line 214) and haze days (e.g. line 231). PM2.5 mass densities are given for the 4 pollution episodes, but what is used to define haze or non-haze days? It would actually be helpful to see time series of the data presented in Figure 2 for the whole campaign for comparison with Fig. 1 and to better justify the selection of the particular phases and episodes. It would also be helpful to mark on the figures, which periods are defined as pollution episodes and which are the phases that are modelled.

Lines 228-230: It states that there was a substantial decline in peroxides during Episode 3, but it looks to me as though their concentrations had already been substantially lower than the previous episodes in the days leading up to episode 3.

Lines 233-4: It states that the peroxides are lower in episode 4 than in the previous

3 episodes, but it looks to me as if their concentrations were already much lower in episode 3.

Regional transport is said to not impact the model results (lines 253-4). It would be good to model episode 4, which was said to be due to regional transport of pollution to demonstrate this – even if this was exhibited by a disagreement between the model and measurements.

Lines 255-262: The data presented in Figure 4 is said to show, for the daytime, an excellent agreement with the modelled data, 1-2 times higher than the measured. Firstly it is not clear if this refers to all the peroxides presented in Fig. 4. Secondly, I would argue that, particularly for MHP and PAA, the difference is often much greater than 2 times. It also needs to be made clear that the model data in Fig. 4 is from the base case, I presume.

In the section 3.2 (lines 267-8) when considering the overestimation of the peroxides concentrations by the model, errors in the source terms are ruled out on the basis that the production terms are constrained by the observations. It needs to be made clear in the Model Description exactly which terms are constrained by the observations. I would also argue that the chemical mechanism for the production of the peroxides is not well understood. Within the MCM many of the rate coefficients of reactions leading to the formation of peroxides are given as generic values (e.g. the rate coefficient for R2), in the absence of species specific measured rates. Furthermore the mechanism used in this modelling study is limited to a subset of reactions containing the reactions of the measured VOCs and subsequent products. Whilst I appreciate this is a sensible modelling approach, the discussion of the results needs to take account of these weaknesses.

The explanation for the night-time discrepancy for Phase 1 (lines 269-280) is attributed to loss on aerosols and the comparison with Beijing refers to haze days and yet Phase 1 is supposedly for the non-haze days. This is very confusing and emphasises my

point above about defining what is meant by a haze day and how this relates to aerosol loading. It states that there is higher aerosol surface area at night concentrations at night. It would be good to present that data. How were the strengths of the missing sinks estimated?

Looking at Fig. 3, and considering the measured-to-model ratio for Phase 2, I am not sure why MHP is singled out, or why 17th June is singled out in line 285. The largest measured values of the peroxides occur on the 16th and the discrepancy for H2O2 looks larger than for MHP. What's more the measured-to-model ratio of 7 for MHP on the 17th looks like it might only be a for short time and at other times the agreement is actually rather good.

Lines 288-9: What evidence is there for biomass burning emissions impacting the composition of the air during phase 2? Are there measurements of acetonitrile, for example? (I see this is covered later with the K+ concentrations, but it would have been good to have mentioned this much earlier).

Lines 291-3: What is also remarkable to me about phase 3 is that there is a lot of variability from day to day in the amounts of the peroxides modelled. In fact this variability is as big, if not bigger than the differences between Phase 3 and Phase 1. On some days the H2O2 and MHP (both modelled and measured) are quite similar for both Phases 1 and 3. I suggest that more could be learnt about the processes affecting the model and measurements by looking at the day to day variability within Phase 3. Did the haze vary much between days? How was the haziness quantified?

Lines 299-305: The text here implies that there were modifications made to the chemical mechanism, with the new reactions given in Table 2. However, it is not clear what they have changed from. New reactions or just new rates?

Lines 321-324: In line 321 it states that the values agree with measurements and then in lines 322-324 say that the additional reactions have marginal impact and are insufficient to match the observations. This is a contradiction as I see it.

When discussing Fig. 3. I would like to see more comment on the shape of the diurnal cycle. The modelled and measured shapes are often very different.

Line 348: The numbers given here are orders of magnitude lower than those for isoprene and trans-2-butene in Fig. 5, so are confusing.

Lines 349-350. This refers to emissions of isoprene and trans-2-butene and yet it is not clear if the model includes emissions. In fact the model description implies that the model is constrained by observed mixing ratio of the VOCs. The model description needs to be much clearer and the text here consistent with the way the model was set up.

Line 368: Whilst I understand that biogenic emissions are not exactly controllable, they can be changed by changes in land use (e.g. different types of vegetation), and policies should consider how such changes could improve or worsen air quality.

It would be helpful to provide a time series of K+ concentrations, for the whole of the campaign to illustrate the periods influenced by biomass burning.

It would also be helpful to annotate Figs. 2 and 3, to show the times of the biomass burning events.

Line 403: How were the strengths of the primary sources calculated?

What is the impact of including primary sources of the peroxides on the chemistry? E.g. does it have an impact on HOX? Are the modelled HOX values comparable to observed values and does the addition of primary sources change this comparison?

Lines 410-13: What other removal processes have been ignored that would lead to underestimating the peroxides, other than heterogeneous loss?

Lines 483-498 Given that OH concentrations and photolysis rates will be reduced in the evenings, the lifetimes calculated will be lower than daytime values. Are the modelled values 24 hour averages? It needs to be clear exactly what you are comparing and if

they are for the same times of day. How sensitive are the modelled relative losses (Fig. 6) to uncertainties in the boundary layer height?

Lines 509-513: Just because the model and measured values of HO2 agree, it doesn't necessarily mean that aerosol uptake of HO2 is insignificant. What is the impact of including H2O2 heterogeneous loss on HO2?

Minor Comments

Line 270, suggest changing "coincides" to "is consistent".

Lines 305-310: Break this sentence up in to smaller sentences.

Line 326: Change "source" to "sources".

Lines 329-336: The English is hard to follow. Break this sentence up in to smaller sentences.

Line 340: Delete "The".

Line 341: Change "that was disregarded" to "not included".

Line 343: Change "seven" to "the seven".

Line 346: Change "shows strong dependence of alkenes" to "shows a strong dependence on alkenes".

Line 351: The logic implied by the word "Besides" does not make sense to me.

Line 353: Change "Such" to "Such a".

Line 358: Change "between two" to "between the two".

Line 360: Delete "the support on the basis of".

Line 364: Change "for the responsible" to "responsible for the".

Line 387: Change "abundance" to "abundances".

Line 387: Change "increased . . .. that" to "were higher than".

Line 388: Change "was" to "is".

Line 395: Change "the concentration of K+ twice more than" to "concentrations of K+ more than twice".

Line 399: Change "the duration" to "durations".

Line 401: Change "observation" to "observations".

Line 404: Change "were" to "are".

Line 421: Change "coincided" to "agreed".

Line 423: Move "relative" to after "PAA".

Line 424: Change "were" to "are".

Line 430: Change "seem to be the" to "are".

Line 424: Change "were" to "are".

Line 438-39: Change "be beneficial" to "help".

Line 444: Change "considerably" to "considerable".

Line 445: Change "made" to "make".

Line 445: Change "of heterogeneous reaction was" to "of the heterogeneous reaction were".

Line 468: Change "heterogeneous" to "the heterogeneous".

Line 489: Change "lifetime" to "lifetimes".

Line 490: Change "whole" to "whole of".

Line 490: Change "was" to "were".

Line 503: Change "referred" to "inferred".

Line 508: Change "are various" to "vary".

Line 516: Delete "is".

Line 517: Change "resulted" to "results".

Line 518: Change "reaction" to "reactions".

Line 532: Change "Wangdu" to "the Wangdu".

Line 536: Change "of" to "of an".

I suggest breaking some of the long paragraphs into shorter ones.

---

## Author Comment (AC1) · 3 Jul 2016

To Reviewer 1

Thank you very much for your time and constructive comments. Here are our responses to your comments.

Overall Comments: This paper presents some interesting ideas and potentially some interesting evidence for direct emissions of peroxides from biomass burning and heterogeneous loss of peroxides. However, it is far from clear how the model was set up, nor how several of the calculated values were derived. This leaves many open questions when it comes to the data analysis. The interpretation appears to focus on some

features of the data and ignoring others, without clear justification. Showing more of the data would better justify the selection of the different phases and episodes and would better illustrate how they differed from each other. This would then strengthen the scientific findings. I have therefore recommended major changes, as I think there are a number of important changes that are required to bring the paper up to a scientifically robust standard to be published in ACP.

Reply: Thanks for your constructive and insightful comments, which indeed help us improve our manuscript. In the revised manuscript, we have provided more evidence for discussion as well as clearer descriptions on the method of modeling according to your comments. In the following we respond to your specific comments.

Specific Comments

In the model description it seems to suggest that the uncertainty in the model is just due to the uncertainty in the observational constraints (lines184-5). What about uncertainties in chemical mechanism - rate coefficients, reaction products, reaction simplifications?

Reply: The uncertainties in chemical mechanism was calculated to be less than 20% according to the method described in Lu et al. (2012).

Dry deposition is a major sink for peroxides, especially $H_2O_2$. What is the uncertainty in the PBL height? Is this included in the total model error?

Reply: Dry deposition is not a major sink for peroxides. The uncertainty in the PBL height was estimated to be 25% according to Xu et al. (2013). It is included in the total model error.

Assuming that data below the D.L. is equal to the D.L. will lead to an overestimation on average (lines 204-5). I would suggest using a value of 0.5 D.L..

Reply: Yes, we have replaced the data below the D.L. with half of the D.L. and modified the Table 1, Fig. 1-3. Please see the revised manuscript.

Lines 211-13: If MHP and PAA being about 20% and 5% of the total peroxides in Wangdu is similar to the results from other rural sites in China, then presumably it could also be stated that with 70% of the peroxides in Wangdu being $H_2O_2$ that Wangdu is also similar to other rural sites in China?

Reply: Yes.

Lines 214-245: It would be helpful to provide more clarity on what is meant by severe pollution episodes (line 214) and haze days (e.g. line 231). $PM_{2.5}$ mass densities are given for the 4 pollution episodes, but what is used to define haze or non-haze days?

Reply: Yes. In the present work, on the basis of the latest national Ambient Air Quality Standards of China (GB3095-2012), the haze pollution episode is defined as the event that a set of continuous days with daily-averaged $PM_{2.5}$ concentration exceeds 75 $\mu$g m$^{-3}$, which has been used to distinguish non-haze and haze episode in the literature (Che et al., 2014; Zhang et al., 2016; Zheng et al., 2015; Zheng et al 2016).

It would actually be helpful to see time series of the data presented in Figure 2 for the whole campaign for comparison with Fig. 1 and to better justify the selection of the particular phases and episodes. It would also be helpful to mark on the figures, which periods are defined as pollution episodes and which are the phases that are modelled.

Reply: Yes. We have modified Fig. 2. Please see the revised manuscript.

Lines 228-230: It states that there was a substantial decline in peroxides during Episode 3, but it looks to me as though their concentrations had already been substantially lower than the previous episodes in the days leading up to episode 3.

Reply: Yes. The mean MHP and PAA levels in the days leading up to Episode 3 (27−28 June) and in Episode 3 were comparable, while the mean $H_2O_2$ level during Episode 3 (0.28 ppbv) declined compared with that on 27−28 June (0.37 ppbv). We have changed the statement from "a substantial decline of $H_2O_2$, MHP and PAA level" to "a substantial decline of $H_2O_2$ level".

Lines 233-4: It states that the peroxides are lower in episode 4 than in the previous 3 episodes, but it looks to me as if their concentrations were already much lower in episode 3.

Reply: Yes. The mean $H_2O_2$ concentration in Episode 3 (0.28 ppbv) was lower than that in Episode 4 (0.32 ppbv), but the mean MHP concentration in Episode 3 (0.12 ppbv) was higher than that in Episode 4 (0.09 ppbv). We have changed the statement from "compared with the other three episodes" to "compared with Episode 1 and Episode 2".

Regional transport is said to not impact the model results (lines 253-4). It would be good to model episode 4, which was said to be due to regional transport of pollution to demonstrate this – even if this was exhibited by a disagreement between the model and measurements.

Reply: Yes, we agree. However, it is unfortunate that the detailed VOC data for episode 4 are unavailable. So we do not simulate the concentrations of atmospheric peroxides during episode 4.

Lines 255-262: The data presented in Figure 4 is said to show, for the daytime, an excellent agreement with the modelled data, 1-2 times higher than the measured. Firstly it is not clear if this refers to all the peroxides presented in Fig. 4. Secondly, I would argue that, particularly for MHP and PAA, the difference is often much greater than 2 times. It also needs to be made clear that the model data in Fig. 4 is from the base case, I presume.

Reply: Yes. We have clarified the text as: "the model base case prediction of $H_2O_2$ level had good performance in the daytime". We have also clarified the title of Fig. 4 with "the model base case" added.

In the section 3.2 (lines 267-8) when considering the overestimation of the peroxides concentrations by the model, errors in the source terms are ruled out on the basis that

the production terms are constrained by the observations. It needs to be made clear in the Model Description exactly which terms are constrained by the observations. I would also argue that the chemical mechanism for the production of the peroxides is not well understood. Within the MCM many of the rate coefficients of reactions leading to the formation of peroxides are given as generic values (e.g. the rate coefficient for R2), in the absence of species specific measured rates. Furthermore the mechanism used in this modelling study is limited to a subset of reactions containing the reactions of the measured VOCs and subsequent products. Whilst I appreciate this is a sensible modelling approach, the discussion of the results needs to take account of these weaknesses.

Reply: Yes. Measurements of NO/$NO_2$, CO, $O_3$, HONO, NMHCs, temperature, pressure and $H_2O$ were used as inputs to constrain the model calculations. The measured and modelled $HO_2$ concentrations at Wangdu site are close to each other. The secondary production terms of atmospheric peroxides including peroxy radical self/cross reactions and the ozonolysis of unsaturated VOCs are constrained by the observations. We have deleted 'by assuming that the chemical mechanisms of atmospheric peroxides are well-understood'.

The explanation for the night-time discrepancy for Phase 1 (lines 269-280) is attributed to loss on aerosols and the comparison with Beijing refers to haze days and yet Phase 1 is supposedly for the non-haze days. This is very confusing and emphasises my point above about defining what is meant by a haze day and how this relates to aerosol loading. It states that there is higher aerosol surface area at night concentrations at night. It would be good to present that data. How were the strengths of the missing sinks estimated?

Reply: Yes, we have presented the mean surface area concentration at daytime and nighttime. Please see the revised manuscript. The strengths of the missing sinks were quantified by the difference between modelled and measured peroxide concentrations.

Looking at Fig. 3, and considering the measured-to-model ratio for Phase 2, I am not sure why MHP is singled out, or why 17th June is singled out in line 285. The largest measured values of the peroxides occur on the 16th and the discrepancy for $H_2O_2$ looks larger than for MHP. What's more the measured-to-model ratio of 7 for MHP on the 17th looks like it might only be a for short time and at other times the agreement is actually rather good.

Reply: Yes. The measured-to-model ratio of 7 for MHP on 17 June was for about 2 hours. We have deleted "the measurement-to-model ratio based on the model case was up to a factor of 7 for MHP on 17 June, which was much higher than the measurement and model errors".

Lines 288-9: What evidence is there for biomass burning emissions impacting the composition of the air during phase 2? Are there measurements of acetonitrile, for example? (I see this is covered later with the $K^+$ concentrations, but it would have been good to have mentioned this much earlier).

Reply: Yes, we have provided a time series for $K^+$ in $PM_{2.5}$ in Fig. 2 and mentioned this in Sect. 3.1. Please see the revised manuscript.

Lines 291-3: What is also remarkable to me about phase 3 is that there is a lot of variability from day to day in the amounts of the peroxides modelled. In fact this variability is as big, if not bigger than the differences between Phase 3 and Phase 1. On some days the $H_2O_2$ and MHP (both modelled and measured) are quite similar for both Phases 1 and 3. I suggest that more could be learnt about the processes affecting the model and measurements by looking at the day to day variability within Phase 3. Did the haze vary much between days? How was the haziness quantified?

Reply: Yes. The haziness was quantified according to the mean $PM_{2.5}$ concentrations over 75 $\mu$g m$^{-3}$. The haze on 1 July and 3 July was more serious with the daily-averaged $PM_{2.5}$ concentrations 1.6 times higher than those on 29 June, 30 June and 2 July. We have added the following statement in the revised manuscript: "The haze

arose on 29 June with the elevated $PM_{2.5}$ concentration. The diffusion condition was poor as the CO concentration was enhanced. The precursors of atmospheric peroxides also accumulated on 29 June and 30 June. The modelled peroxide concentrations over 10 times higher than the measured peroxide concentrations. On 1 July and 3 July, the daily-averaged $PM_{2.5}$ concentration was 1.6 times higher than those on 29 June and 30 June. However, the photolysis frequencies and the PBL height on 1 July and 3 July were about half of those on 29 June and 30 June, which weakened the secondary formation of atmospheric peroxides and strengthened the loss of atmospheric peroxides via dry deposition. Although the haze on 1 July and 3 July was more serious than that on 29 June and 30 June, the ratios of modelled to measured peroxide concentrations on 1 July and 3 July were much lower than those on 29 June and 30 June.".

Lines 299-305: The text here implies that there were modifications made to the chemical mechanism, with the new reactions given in Table 2. However, it is not clear what they have changed from. New reactions or just new rates?

Reply: For $CH_3O_2$ related chemistry, the new reaction between $CH_3O_2$ radicals and $HO_2$ radicals was incorporated in our box model. For $CH_3C(O)O_2$ related chemistry, only new rate constants were employed in our box model.

Lines 321-324: In line 321 it states that the values agree with measurements and then in lines 322-324 say that the additional reactions have marginal impact and are insufficient to match the observations. This is a contradiction as I see it.

Reply: Yes. We have deleted "Moreover. . .. errors".

When discussing Fig. 3. I would like to see more comment on the shape of the diurnal cycle. The modelled and measured shapes are often very different.

Reply: Yes. With the inclusion of heterogeneous uptake of atmospheric peroxides, the modelled and measured shapes in the Phase II are different due to the lack of possible primary emission from biomass burning in our box model, while the modelled

and measured shapes in the Phase I and the Phase III are similar and can be explained by the model-measurement uncertainty.

Line 348: The numbers given here are orders of magnitude lower than those for isoprene and trans-2-butene in Fig. 5, so are confusing.

Reply: Yes. We have deleted "In. . .. seen".

Lines 349-350. This refers to emissions of isoprene and trans-2-butene and yet it is not clear if the model includes emissions. In fact the model description implies that the model is constrained by observed mixing ratio of the VOCs. The model description needs to be much clearer and the text here consistent with the way the model was set up.

Reply: Yes. The model does not include emissions. We have changed the statement from "isoprene from the local biogenic emission and trans-2-butene from the anthropogenic emission diversity" to "isoprene and trans-2-butene".

Line 368: Whilst I understand that biogenic emissions are not exactly controllable, they can be changed by changes in land use (e.g. different types of vegetation), and policies should consider how such changes could improve or worsen air quality.

Reply: Yes, we agree. We have clarified the text as: "It is recommended to take measures for vehicle emission control and land use management (e.g. modifying the amount and types of vegetation) in order to mitigate the pollution of atmospheric peroxides in the NCP and hence alleviate their potential harmful effects on air quality, human health and ecosystem".

It would be helpful to provide a time series of $K^+$ concentrations, for the whole of the campaign to illustrate the periods influenced by biomass burning.

Reply: Yes. We have provided a time series for $K^+$ in $PM_{2.5}$ in Fig. 2. In addition, we have redefined the biomass burning Event II as 16:00–19:00 on 16 June, for it exhibited more excellent correlation with $K^+$. We have also modified Table 3. Please

see the revised manuscript.

It would also be helpful to annotate Figs. 2 and 3, to show the times of the biomass burning events.

Reply: Yes. We have modified Fig. 2-3. Please see the revised manuscript.

Line 403: How were the strengths of the primary sources calculated?

Reply: The strengths of the primary sources were quantified by the difference between modelled and measured peroxide concentrations.

What is the impact of including primary sources of the peroxides on the chemistry? E.g. does it have an impact on $HO_x$? Are the modelled $HO_x$ values comparable to observed values and does the addition of primary sources change this comparison?

Reply: The impact of primary sources of the peroxides on $HO_x$ radicals was marginal as the primary production of $HO_x$ radicals was dominated by the photolysis of HONO, $O_3$, HCHO and dicarbonyls. The change of modelled $HO_x$ values was almost negligible with the addition of primary sources in our box model. The observed OH concentration was underestimated, while the observed $HO_2$ concentration was reproduced by the model. More data analysis on the $HO_x$ chemistry can be obtained from Tan et al. (2016).

Lines 410-13: What other removal processes have been ignored that would lead to underestimating the peroxides, other than heterogeneous loss?

Reply: We think that there is no removal pathway other than heterogeneous loss that would lead to the overestimation of atmospheric peroxides.

Lines 483-498: Given that OH concentrations and photolysis rates will be reduced in the evenings, the lifetimes calculated will be lower than daytime values. Are the modelled values 24 hour averages? It needs to be clear exactly what you are comparing and if they are for the same times of day. How sensitive are the modelled relative losses

(Fig. 6) to uncertainties in the boundary layer height?

Reply: Yes. The modelled values were 24 hour averages. The comparison was done for the same times of day. The sensitivity of modelled relative loss of dry deposition to the uncertainties in the planetary boundary layer height was low, for instance, the contribution of dry deposition to the loss of $H_2O_2$ in Phase I decreased from 17% to 10% with the PBL height doubled. Heterogeneous uptake by aerosols was still the predominant sink for atmospheric peroxides.

Lines 509-513: Just because the model and measured values of $HO_2$ agree, it doesn't necessarily mean that aerosol uptake of $HO_2$ is insignificant. What is the impact of including $H_2O_2$ heterogeneous loss on $HO_2$?

Reply: Yes. We have clarified the text as: "the impact of aerosol uptake of $HO_2$ radicals on the concentration of atmospheric peroxides is insignificant". The impact of $H_2O_2$ heterogeneous loss on $HO_2$ radical concentrations was marginal.

Minor Comments

Line 270, suggest changing "coincides" to "is consistent".

Reply: Yes.

Lines 305-310: Break this sentence up in to smaller sentences.

Reply: Yes, we have broken this sentence into smaller ones.

Line 326: Change "source" to "sources".

Reply: Yes.

Lines 329-336: The English is hard to follow. Break this sentence up in to smaller sentences.

Reply: Yes, we have broken this sentence into smaller ones.

Line 340: Delete "The".

Reply: Yes.

Line 341: Change "that was disregarded" to "not included".

Reply: Yes.

Line 343: Change "seven" to "the seven".

Reply: Yes.

Line 346: Change "shows strong dependence of alkenes" to "shows a strong dependence on alkenes".

Reply: Yes.

Line 351: The logic implied by the word "Besides" does not make sense to me.

Reply: Yes, we have replaced "Besides" with "Moreover".

Line 353: Change "Such" to "Such a".

Reply: Yes.

Line 358: Change "between two" to "between the two".

Reply: Yes.

Line 360: Delete "the support on the basis of".

Reply: Yes.

Line 364: Change "for the responsible" to "responsible for the".

Reply: Yes.

Line 387: Change "abundance" to "abundances".

Reply: Yes.

Line 387: Change "increased . . .. that" to "were higher than".

Reply: Yes.

Line 388: Change "was" to "is".

Reply: Yes.

Line 395: Change "the concentration of K$^+$ twice more than" to "concentrations of K$^+$ more than twice".

Reply: Yes.

Line 399: Change "the duration" to "durations".

Reply: Yes.

Line 401: Change "observation" to "observations".

Reply: Yes.

Line 404: Change "were" to "are".

Reply: Yes.

Line 421: Change "coincided" to "agreed".

Reply: Yes.

Line 423: Move "relative" to after "PAA".

Reply: Yes.

Line 424: Change "were" to "are".

Reply: Yes.

Line 430: Change "seem to be the" to "are".

Reply: Yes.

Line 424: Change "were" to "are".

Reply: Yes.

Line 438-39: Change "be beneficial" to "help".

Reply: Yes.

Line 444: Change "considerably" to "considerable".

Reply: Yes.

Line 445: Change "made" to "make".

Reply: Yes.

Line 445: Change "of heterogeneous reaction was" to "of the heterogeneous reaction were".

Reply: Yes.

Line 468: Change "heterogeneous" to "the heterogeneous".

Reply: Yes.

Line 489: Change "lifetime" to "lifetimes".

Reply: Yes.

Line 490: Change "whole" to "whole of".

Reply: Yes.

Line 490: Change "was" to "were".

Reply: Yes.

Line 503: Change "referred" to "inferred".

Reply: Yes.

Line 508: Change "are various" to "vary".

Reply: Yes.

Line 516: Delete "is".

Reply: Yes.

Line 517: Change "resulted" to "results".

Reply: Yes.

Line 518: Change "reaction" to "reactions".

Reply: Yes.

Line 532: Change "Wangdu" to "the Wangdu".

Reply: Yes.

Line 536: Change "of" to "of an".

Reply: Yes.

I suggest breaking some of the long paragraphs into shorter ones.

Reply: Yes, we have broken the long paragraphs into shorter ones. Please see the revised manuscript.

References

Che, H., Xia, X., Zhu, J., Li, Z., Dubovik, O., Holben, B., Goloub, P., Chen, H., Estelles, V., Cuevas-Agulló, E., Blarel, L., Wang, H., Zhao, H., Zhang, X., Wang, Y., Sun, J., Tao, R., Zhang, X. and Shi, G.: Column aerosol optical properties and aerosol radiative forcing during a serious haze-fog month over North China Plain in 2013 based on ground-based sunphotometer measurements, Atmos. Chem. Phys., 14, 2125−2138, 2014.

Lu, K. D., Rohrer, F., Holland, F., Fuchs, H., Bohn, B., Brauers, T., Chang, C. C., Häseler, R., Hu, M., Kita, K., Kondo, Y., Li, X., Lou, S. R., Nehr, S., Shao, M., Zeng, L.

M., Wahner, A., Zhang, Y. H. and Hofzumahaus, A.: Observation and modelling of OH and HO$_2$ concentrations in the Pearl River Delta 2006: a missing OH source in a VOC rich atmosphere, Atmos. Chem. Phys., 12, 1541$-$1569, 2012.

Tan, Z. F., Fuchs, H., Lu, K. D., Bohn, B., Broch, S., Haeseler, R., Hofzumahaus, A., Holland, F., Li, X., Liu, Y., Rohrer, F., Shao, M., Wang, B. L., Wang, M., Wu, Y. S., Zeng, L. M., Wahner, A. and Zhang, Y. H.: Observation and modelling of the OH, HO$_2$ and RO$_2$ radicals at a rural site (Wangdu) in the North China Plain in summer 2014, Geophysical Research Abstracts, pp. EGU2016$-$5459, 2016.

Xu, W. Y., Zhao, C. S., Ran, L., Deng, Z. Z., Ma, N., Liu, P. F., Lin, W. L., Yan, P., and Xu, X. B.: A new approach to estimate pollutant emissions based on trajectory modeling and its application in the North China Plain, Atmos. Environ., 71, 75$-$83, 2013.

Zhang, Y., Huang, W., Cai, T. Q., Fang, D. Q., Wang, Y. Q., Song, J., Hu, M. and Zhang, Y. X.: Concentrations and chemical compositions of fine particles (PM$_{2.5}$) during haze and non-haze days in Beijing, Atmos. Res., 174, 62$-$69, 2016.

Zheng, G. J., Duan, F. K., Ma, Y. L., Zhang, Q., Huang, T., Kimoto, T. K., Cheng, Y. F., Su, H. and He, K. B.: Episode-based evolution pattern analysis of haze pollution: method development and results from Beijing, China, Environ. Sci. Technol., 50, 4632$-$4641, 2016.

Zheng, G. J., Duan, F. K., Su, H., Ma, Y. L., Cheng, Y., Zheng, B., Zhang, Q., Huang, T., Kimoto, T., Chang, D., Poschl, U., Cheng, Y. F. and He, K. B.: Exploring the severe winter haze in Beijing: the impact of synoptic weather, regional transport and heterogeneous reactions, Atmos. Chem. Phys., 15, 2969$-$2983, 2015.

---

## Author Comment (AC2) · 3 Jul 2016

Thank you very much for your careful review.
* * *

---

## Author Response (AR2)

July 25, 2016

Dear Prof. Heard,

Enclosed please find our revised manuscript acp-2016-292 entitled "*Observation of atmospheric peroxides during Wangdu Campaign 2014 at a rural site in the North China Plain*". We have completed the revisions according to your comments. We have uploaded all of the materials to the ACP website.

Many thanks for your comments and time.

Sincerely yours,

Zhongming Chen and co-authors

Comments to the Author:

Thank you for the responses to the comments of reviewer (1).

There are a few things I would like you to expand on a little. Mainly it is when you give a response, but the MS does not seem to have been modified to reflect the response. See below.

The reviewer asks:
In the model description it seems to suggest that the uncertainty in the model is just due to the uncertainty in the observational constraints (linee184-5). What about uncertainties in chemical mechanism - rate coefficients, reaction products, reaction simplifications?

Your reply to this:
"The uncertainties in chemical mechanism was calculated to be less than 20% according to the method described in Lu et al. (2012)."

This is somewhat vague. Can you please indicate which parts of the chemical mechanism are likely to have the largest uncertainties and give a little more detail on those areas of the mechanism which contribute most to the overall 20% that Lu et al. report.

Reply: Yes. The largest uncertainties in chemical mechanism are possibly from the rate constants of peroxy radical self/cross reaction related to the secondary formation of $H_2O_2$, MHP and PAA. The uncertainties in the rate constants of the reaction between $HO_2$ radicals, the reaction between $CH_3O_2$ radicals and $HO_2$ radicals and the reaction between $CH_3C(O)O_2$ radicals and $HO_2$ radicals are calculated to be ~15%

with an accuracy of 30% on the rate constant in MCMv3.3.1. The detail on the areas of mechanism that contribute most to the overall uncertainty in Lu et al. (2012) is not available as we just employed the method described in Lu et al. (2012) to estimate our model uncertainty.

Can you please indicate the fraction of peroxides which are lost by dry deposition. You state it is not major, whereas the referee states that dry deposition is major - can you please provide some evidence that dry deposition is minor.

Reply: Yes. Dry deposition is the marginal removal pathway that contributed less than 20% to the destruction of $H_2O_2$, MHP and PAA, as shown in Fig. 6 in the MS. The simulated mean lifetime of $H_2O_2$, MHP and PAA between 18:00 and 24:00 LT in the Phase I can be about 14 h, 16 h and 21 h, respectively, supposing that the loss of $H_2O_2$, MHP and PAA is merely due to photolysis, OH-initiated reaction and dry deposition, which is strongly overestimated compared with the mean lifetime of $H_2O_2$, MHP and PAA obtained from the field observation between 18:00 and 24:00 LT in the Phase I (<6 h). Heterogeneous uptake by aerosols was the predominant sink for atmospheric peroxides during Wangdu Campaign 2014.

Referee said:
Lines 211-13: If MHP and PAA being about 20% and 5% of the total peroxides in Wangdu is similar to the results from other rural sites in China, then presumably it could also be stated that with 70% of the peroxides in Wangdu being $H_2O_2$ that Wangdu is also similar to other rural sites in China?

Your reply is "yes" but can you make sure that this is stated in the MS - you did not say you had done this.

Reply: Yes. We have stated this in the MS as "$H_2O_2$ accounted for ~70% of total detected peroxides ($H_2O_2$ + MHP + PAA) similar to those determined at other rural sites in China".

Please state in the paper the criteria for when there is a "haze day". You state this nicely in the reply but not clear if stated in the MS - please ensure that it is.
It is very important that a clear definition of a haze day is contained in the MS.

Reply: Yes. We have stated this in the MS as "the haze day is defined as a day with daily-averaged $PM_{2.5}$ concentration over 75 $\mu g\ m^{-3}$".

The referee states:
Lines 299-305: The text here implies that there were modifications made to the chemical mechanism, with the new reactions given in Table 2. However, it is not clear what they have changed from. New reactions or just new rates?

In your reply you say the new reaction between $CH_3O_2$ and $HO_2$ was incorporated into your model. Please state clearly in the text exactly what you have done - state the old rate constants and what you have changed them to.

Reply: Sorry. This is a mistake in our response to reviewer (1). For $CH_3O_2$ related chemistry, the new reaction between $CH_3O_2$ radicals and OH radicals was incorporated in our box model. Please see the former revised manuscript.

The referee says:
When discussing Fig. 3. I would like to see more comment on the shape of the diurnal cycle. The modelled and measured shapes are often very different.

You reply to this, but you do not seem to say you have changed the MS for this. Please put more comment on the shape of the diurnal cycle in the MS as the referee asks.

Reply: Yes. We have stated this in Sect. 3.2 in the MS as " the calculated values in the model base case showed a general tendency to strongly overestimate the observed values (Fig. 3). The modelled and measured shapes of the diurnal cycle of atmospheric peroxides were different", "with the inclusion of heterogeneous reactions on aerosol particles, the simulated concentrations of atmospheric peroxides were apparently improved and the modelled shape of the diurnal cycle of $H_2O_2$ was closer to the measured shape" and in Sect. 3.4 in the MS as "The modelled and measured shape of the diurnal cycle of $H_2O_2$ in the Phase I and the Phase III are similar". Please see the revised manuscript.

The referee says:
What is the impact of including primary sources of the peroxides on the chemistry? E.g. does it have an impact on $HO_x$? Are the modelled $HO_x$ values comparable to observed values and does the addition of primary sources change this comparison?

You say the impact on $HO_x$ is marginal. Can you please give a precise % value for this and put this in the MS.

Reply: Yes. We have stated this in the MS as "The impact of primary sources of the peroxides on $HO_x$ radicals was limited with the increase of OH radicals not more than 10% and the increase of $HO_2$ radicals not more than 5%".

Referee question and response:
Lines 483-498: Given that OH concentrations and photolysis rates will be reduced in the evenings, the lifetimes calculated will be lower than daytime values. Are the modelled values 24 hour averages? It needs to be clear exactly what you are comparing and if they are for the same times of day. How sensitive are the modelled relative losses (Fig. 6) to uncertainties in the boundary layer height?

Reply: Yes. The modelled values were 24 hour averages. The comparison was done for the same times of day. The sensitivity of modelled relative loss of dry deposition to the uncertainties in the planetary boundary layer height was low, for instance, the contribution of dry deposition to the loss of $H_2O_2$ in Phase I decreased from 17% to 10% with the PBL height doubled. Heterogeneous uptake by aerosols was still the predominant sink for atmospheric peroxides.

Thanks for detailed response, but please make sure this is all contained in the revised MS.

Reply: Yes. We have stated this in the MS as "Although dry deposition is thought to dominate the atmospheric lifetime of peroxides in previous studies (Reeves and Penkett, 2003), its role in the lifetime of atmospheric peroxides is insignificant during Wangdu Campaign 2014. The sensitivity of modelled relative loss of dry deposition to the uncertainties in the planetary boundary layer height was low as the contribution of dry deposition to the loss of $H_2O_2$ in Phase I decreased no more than 10% with the PBL height doubled". We have also clarified the statement further by modifying "the lifetime" to "the daily-averaged lifetime". Please see the revised manuscript.

Fig 2, 3 replace "the grey shade" with "the grey shaded area...". Same for "orange shade"

Reply: Yes. We have changed "shade" to "shaded area".

References

[revised manuscript text omitted]